# Improving the Management of a Semi-Arid Agricultural Ecosystem through Digital Mapping of Soil Properties: The Case of Salamanca (Spain)

**Marco Criado [1],\*** , **Antonio Martínez-Graña [2]** , **Fernando Santos-Francés [1]** and **Leticia Merchán [1]**

[1] Department of Soil Sciences, Faculty of Agricultural and Environmental Sciences, University of Salamanca, Avenue Filiberto Villalobos, 119, 37007 Salamanca, Spain; fsantos@usal.es (F.S.-F.); leticiamerchan@usal.es (L.M.)

[2] Department of Geology, Faculty of Sciences, University of Salamanca, Plaza de la Merced s/n., 37008 Salamanca, Spain; amgranna@usal.es

\* Correspondence: marcocn@usal.es

**Abstract:** Soil protection and the increase and intensification of agricultural production require detailed knowledge of soil properties and their variability. On the other hand, the complexity associated with traditional soil mapping processes can lead to the implementation of inappropriate agricultural practices that degrade this resource. Therefore, it is necessary to use mapping techniques to provide more detailed information to farmers and managers. In this study, the geostatistical technique ordinary kriging was used to map the distribution of the most important edaphic properties (texture, nutrients content -N, P, K-, pH, organic carbon, water retention, COLE, carbonate content, and cation exchange capacity) from known sampled points, which allows inferring the value and distribution of the different edaphic parameters studied along the agricultural fields. The results obtained show after validation that the analysis of semivariograms is suitable for evaluating the distribution of the main soil parameters on a large scale, since it faithfully reflects their distribution and makes the ordinary kriging tool a suitable method for optimizing the resources available in soil mapping processes. In addition, the knowledge of these distributions made it possible to establish different recommendations for improving the management of the agricultural ecosystem, which will guarantee a higher agricultural yield as well as a better protection of the analyzed soils.

**Keywords:** agrarian ecosystem; GIS; geostatistics; kriging; soil mapping

## 1. Introduction

Agriculture is currently facing multiple challenges with high global implications, especially the food supply of a growing population [1]. Although agricultural knowledge and techniques have experienced a high development in the last decades that has contributed to meet food demands [2], other problems have not been solved and put agricultural activity at risk in the current stage of productive intensification [3,4]. Poor agricultural practices and lack of agronomic information can lead to soil erosion [5], contribute to global warming [6], affect biodiversity [7], and cause contamination of agricultural land and water [8,9]. Specifically, the lack of information about soils is of concern, which is usually limited to isolated analytics that do not reflect soil variability and, therefore, the farmer does not know the different soil requirements in each location [10]. In-depth knowledge of soil characteristics is necessary to achieve sustainable agriculture that avoids malpractices [11].

Digital soil mapping (DSM) using statistical inference and GIS is a technique that allows researchers to accurately interpolate spatial patterns of soil properties [12]. The use of the geostatistical interpolation technique also reduces the costs of field sampling and laboratory analysis and allows the soil mapping process to be accelerated [13]. However, the reliability of spatial variability maps depends on adequate sampling data (such as the

number of samples or the distance between them) and the accuracy of spatial interpolation [11]. There are several interpolation methods, including deterministic and stochastic ones, well compiled by Myers [14]. Among them, kriging has proven to be robust enough to estimate values at unsampled locations from the study of specific sampled locations. Moreover, kriging provides the best unbiased linear estimates and information on the estimation error distribution and shows strong statistical advantages over other geostatistical methods (Wang et al., 2012). Several authors already employed this technique in predictive soil mapping, studying different factors: various edaphic parameters [15,16]; nutrient distribution [17], erosion estimation [18]; organic carbon [19–23]; organic matter [24]; heavy metals [25]; chemical properties with importance in agriculture [10]; or soil quality indices [26].

Despite the good adaptation of these interpolation techniques to predictive soil mapping, there are no studies that represent the variability of the main edaphic properties in the study sector. In practice, this translates into the proliferation of agricultural practices inadequate to the needs of the soils. Therefore, the objectives of this work are: (1) to predict the distribution of soil properties that have a greater relevance in agricultural development from the sampled soils and the geostatistical technique of ordinary kriging interpolation; and (2) to improve agronomic knowledge of the soils in the study area and to promote increased crop yields through the use of sustainable agricultural management practices.

## 2. Materials and Methods

### 2.1. Study Area

The study area is located in the west of Spain and southwest of the region of Castilla y León, more specifically, in the northeast of the province of Salamanca (Figure 1). It covers a total area of 770 km$^2$ with its epicenter in the city of Salamanca. In the center and north is the region of La Armuña, famous for its agricultural products, and in the south the Campo Charro, characterized by extensive livestock farming. Geographically, the region studied is located within the Spanish Northern Plateau, with a flat to gently undulating topography, as evidenced by the small difference in altitude (182 m.) between the highest peaks (Los Montalvos mountain, 942 m.a.s.l., modeled on a synclinoric syncline) and the highest peaks (Los Montalvos, 942 m.a.s.l., modeled on a synclinorium formed by Armorican quartzite and slate of Ordovician-Silurian age) and minimum (corresponding to the fertile lowlands of the Tormes River, on alluvial deposits of the Pleistocene-Holocene with heights of 760 m.), predominating in the intermediate heights the plains carved on Palaeogene and Neogene horizontal sediments [27].

The climate is characterized by long, cold winters and hot, dry summers, with an average annual precipitation and temperature of 400 mm and 12 °C, respectively, and precipitation is concentrated mainly in the winter period (between October and March). The average temperature in the summer months (June, July, and August) is 21 °C and the average temperature in the winter months is 6 °C [27]. These temperatures define a climate of short, relatively cool summers and long, harsh winters, with the frost period lasting from mid-October to mid-May. The soils of this region have a xeric humidity regime and a mesic temperature regime [27].

The vegetation present in the studied area is scarce due to the high anthropization of the area, derived mainly from the intense agricultural activity that has led to the deforestation of the climactic Mediterranean forest for centuries. Currently, together with the predominant crops, small redoubts of holm oak groves and holm oak pastures, as well as areas with natural grasslands and riparian vegetation [28,29] constitute the general cover of the environment.

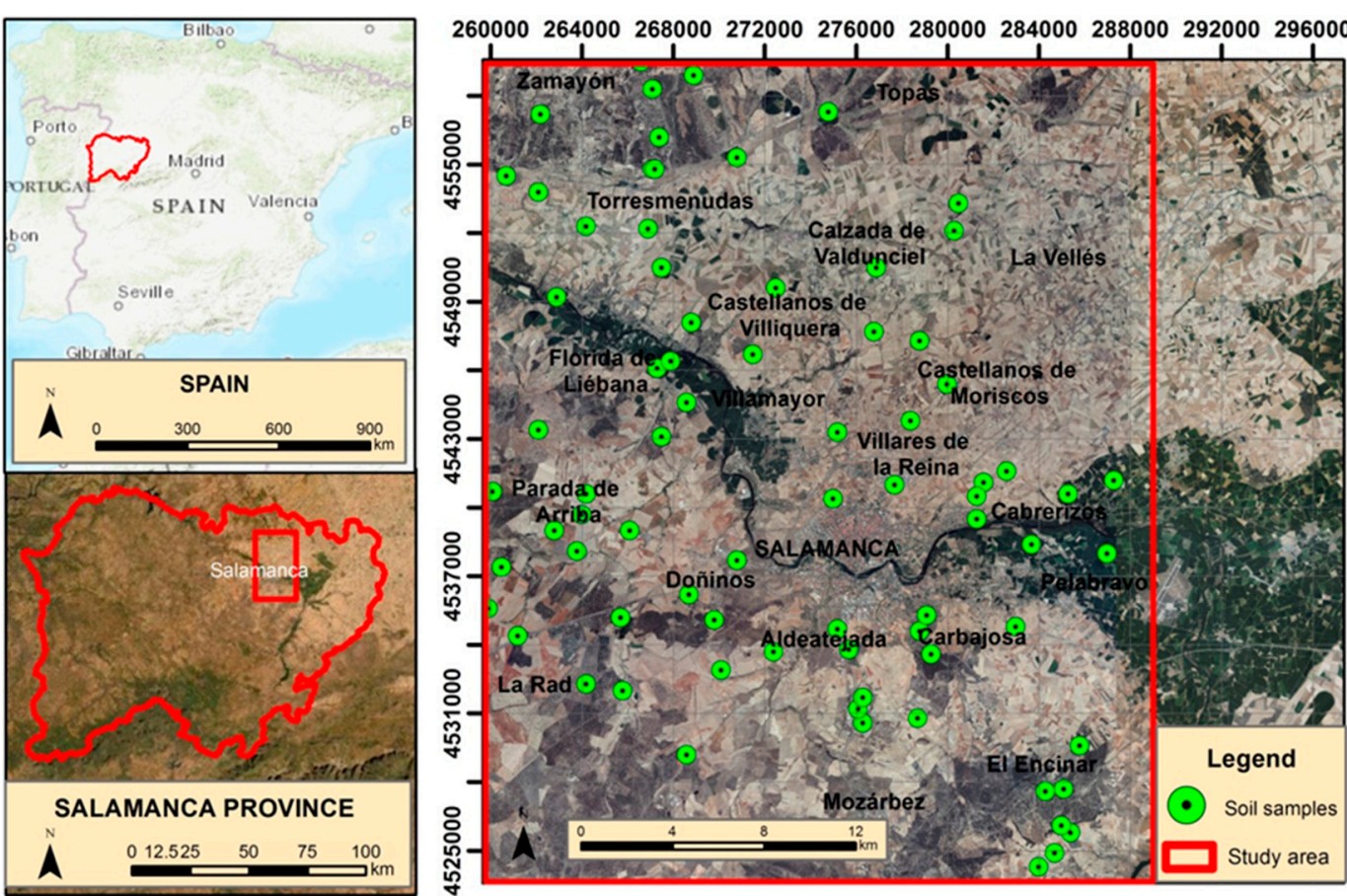

**Figure 1.** Location of the study area and soils sampled.

The soils of the study area [27], classified according to the soil taxonomy method World Reference Base for Soil Resources of the FAO [30], are mostly Luvisols and Cambisols and, to a lesser extent, Fluvisols, Arenosols, Regosols, and Leptosols. This region is dominated by plains tilled over Cenozoic sediments, with the presence of powerful and fertile soils, which have allowed the establishment of high-yield rain-fed agriculture since Roman times, which considered this region as "the breadbasket of Spain" [31]. The agricultural management carried out in this sector is linked to the traditional practices of the area, with deep ploughing with soil turning (cross ploughing with subsoilers or moldboards) and subsequent shallower ploughing with cultivators or disc harrows. These practices are also associated with the indiscriminate application of fertilizers and phytosanitary products. The main crops grown are rain-fed winter cereals, especially wheat, barley, and rye, some leguminous plants such as lentils and chickpeas, and different combinations of varieties suitable for fodder for livestock feed. Irrigated agriculture predominates in the soils of the Tormes valley, with maize and sugar beet as the main crops.

### 2.2. Sampling, Soil Analysis, and Selection of Soil Properties

For the general characterization of the soils of the region, 76 soils distributed by the different lithologies and physiographic positions were studied. Once the different sampling points were located, a sample was taken from the superficial horizon (0–25 cm), since it is the one that corresponds to the arable layer and, therefore, has the greatest importance in the development of crops. Subsequently, they were air-dried, crumbled, and sieved through a 2 mm sieve, before an analysis of the complete physicochemical properties was carried out.

From the total number of properties analyzed, available for the entire estimated area, a representative set was chosen for the study to include those parameters of greatest importance for determining the agricultural conditions of the soil. The choice of parameters, moreover, coincides with the opinions of several authors due to their influence on soil fertility, nutrient supply, root growth, and soil porosity [32–38]. The properties studied are 12: nitrogen (N), assimilable phosphorus (P), and assimilable potassium (K), due to their importance in soil fertility, and texture (sand and clay content), cation exchange capacity (CEC), organic carbon (OC), pH, carbonate content (CaCO$_3$), COLE and water retention at 33 and 1500 kPa (GWC) being the most relevant properties in soils as demonstrated in previous studies [26]. The methods used in the determination of the 12 selected properties are listed in (Table 1).

**Table 1.** Methods used for the analysis of the studied properties.

| Parameter | Units | Method |
| --- | --- | --- |
| Granulometric analysis (sand and clay) | % | Robinson pipette [39] |
| Organic carbon (OC) | % | Dichromate oxidation [40] |
| Water retention at 33 and 1500 kPa (GWC) | % | Pressure membrane [41] |
| Cation exchange capacity (CEC) | cmol kg$^{-1}$ | Ammonium acetate at pH 7 [39] |
| Nitrogen (N) | % | Kjeldahl N [42] |
| Available Phosphorus (P) | mg 100 g$^{-1}$ | Bray 1 [43]/Olsen [44] |
| Available Potassium (K) | mg 100 g$^{-1}$ | Ammonium acetate at pH 7 [39] |
| pH | - | Potentiometric method (1:1 -soil-water-) |
| CaCO$_3$ equivalent (CaCO$_3$) | % | Bernard calcimeter [39] |
| COLE | mm cm$^{-1}$ | Membrana de Richards [39]) |

*2.3. Geostatistical Analysis*

Initially, descriptive statistics were calculated for the properties studied (mean, median, standard deviation, maximum, minimum, kurtosis, and coefficient of variation) to broaden and complement the knowledge of the soils in the environment, for which the SPSS v.25 statistical software was used. After this, from the sample data, we proceeded to perform the interpolation, for which the ordinary kriging method was used.

Ordinary kriging is a linear geostatistical interpolation technique that performs the estimates based on the weighted sums of the sampled point values, and it is well detailed in other works [45–47]. Previously, the normality of the data for all parameters studied was studied using SPSS v.25 statistical software using the visual method (histograms, box-plots, and Q-Q plots). This was refuted by performing the Kolmogorov–Smirnov test at 0.01% probability. The kriging method uses semivariance to estimate the structure of the spatial distribution of soil properties [48]. Modeling and semivariogram estimation are essential for structural analysis and spatial interpolation [10], so the main geostatistical parameters were studied for each property: model type, rank, nugget, structural, threshold, and range. In addition, the spatial dependence (Sp. D), also known as RD (ratio of dependency), of the soil parameters was determined from the relationship between nugget and threshold variations [49]. To ensure spatial dependency, as a rule of thumb, the sampling interval (lag) should be less than half of the range of spatial variation. If the ratio between nugget and Sill (Sp. D) is less than 0.25, the variance has a strong spatial dependence, if the ratio ranges between 0.25 and 0.75, the variance has a moderate spatial dependence, and if it is greater than 0.75, it is considered as slight [46,50].

To perform the interpolation, the dbf file was inserted in ArcMap with the coordinates and sample data related to each property in the 76 profiles studied, the digital layers with the estimate of the distribution of each property were elaborated using the ordinary kriging method and the Geostatistical Analyst module of ArcMap v.10.5 (ESRI). For each parameter, 5 types of estimated distribution were established: very high, high, moderate, low, and very low. For this, the range of each property was divided by the number of desired intervals (5), and the result was used as the width of each interval. Adding this value to the lowest

value of the corresponding property yielded the upper limit of the first interval, and so on, until the upper range of the property was reached. These subdivisions do not indicate a suitability or lack for a given property, but rather a lesser or greater separation from the regional mean.

Finally, the interpolation results were checked using cross-validation [51], for which the Geostatistical Analyst extension of ArcMap is also used. This method consists of eliminating a data location and then predicting the values at that point using the data from the other locations, so that the predicted value is compared with the observed value. Derived from this cross-validation, the mean standardized error (MSE), the root mean square error (RMSE), usually the most commonly used, and the root mean square standardized error (RMSSE) were estimated to verify the adequacy of the accuracy of the kriging tool. A value of MSE close to zero indicates that the interpolation method is unbiased. If the RMSE value is close to the standard deviation of the data, then the model has made an adequate prediction. The RMSSE should be close to one if the standard errors of prediction are valid and if it is greater than one, it is underestimating variability in its predictions, while if it is less than one, it is overestimating variability [52,53].

## 3. Results and Discussion

### 3.1. Descriptive Statistics for Soil Parameters

The compilation of the main descriptive statistics for each edaphic parameter is presented in Table 2. The coefficient of variation reflects the variability of the edaphic properties. It is observed that the CV data are high, which means that there is great edaphic diversity in the sampled area (linked to the lithological and geomorphological diversity of the sector), which is in line with what was observed during the field campaign and in other previous works [26,27]. In addition, the study of the normality of the data determined that the parameters sand, pH, and water retention at 33 kPa showed normal distributions. For the rest of the parameters, the logarithmic transformation was required in order to perform the interpolation. Table 3 shows the rating scales that have been made from the data obtained from the soil profiles studied in this report.

**Table 2.** Descriptive statistics of the edaphic studied parameters.

| Parameters | Mean | Median | St. Desv. | Max. | Min | Kurtosis | CV |
|---|---|---|---|---|---|---|---|
| Sand | 58.19 | 58.48 | 19.40 | 95.34 | 1.04 | −0.88 | 33.34 |
| Clay | 16.76 | 13.29 | 13.59 | 76.11 | 0.67 | −0.11 | 81.09 |
| OC | 1.47 | 1.47 | 1.32 | 7.32 | 0.25 | 24.77 | 89.80 |
| N | 0.10 | 0.07 | 0.10 | 0.61 | 0.01 | 15.76 | 100.0 |
| P | 4.23 | 3.12 | 3.55 | 15.2 | 0.50 | 1.61 | 83.93 |
| K | 10.06 | 8.95 | 7.42 | 58.4 | 3.00 | 17.87 | 73.76 |
| $GWC_{33kPa}$ | 16.70 | 16.19 | 8.67 | 54.61 | 4.36 | 0.06 | 51.92 |
| $GWC_{1500kPa}$ | 7.89 | 6.29 | 6.02 | 32.21 | 0.96 | 0.56 | 76.30 |
| COLE | 0.03 | 0.01 | 0.04 | 0.26 | 0.00 | 4.09 | 133.33 |
| pH | 5.91 | 5.85 | 0.86 | 7.9 | 4.30 | −1.06 | 14.55 |
| CEC | 11.82 | 9.39 | 8.84 | 46.05 | 1.74 | 1.05 | 74.79 |
| $CaCO_3$ | 1.37 | 0.17 | 3.13 | 24.17 | 0.00 | 14.07 | 228.46 |

### 3.2. Soil Properties Maps and Agricultural Management Recommendations

It is well known that plant development is closely related to a series of soil characteristics, such as texture, nutrient content, degree of saturation in bases, cation exchange capacity, organic matter content, salinity, etc. Therefore, these factors are the ones that have been used as a basis for evaluating and mapping soil fertility, except for salinity, because there are no saline soils in the region studied (the average conductivity of the saturation extract is very small −0.83 dSm$^{-1}$-). The results of the interpolations allow us to know the distribution of the main edaphic parameters in the soils of Salamanca and its

surroundings. From this information, more efficient agricultural practices can be developed and recommendations can be made to local farmers [10,54,55].

**Table 3.** Established values for each of the property ranges.

| Property | Classes | | | | |
|---|---|---|---|---|---|
| | I(Very Low) | II(Low) | III(Moderate) | IV(High) | V(Very High) |
| Sand | 1.04–19.90 | 19.91–38.76 | 38.77–57.62 | 57.63–76.48 | 76.49–95.34 |
| Clay | 0.67–15.76 | 15.77–30.85 | 30.86–45.93 | 45.94–61.02 | 61.03–76.11 |
| OC | 0.25–1.66 | 1.67–3.08 | 3.09–4.49 | 4.50–5.91 | 5.92–7.32 |
| N | 0.014–0.134 | 0.136–0.254 | 0.255–0.374 | 0.375–0.494 | 0.495–0.614 |
| P | 0.5–3.4 | 3.5–6.4 | 6.5–9.3 | 9.4–12.3 | 12.4–15.2 |
| K | 3.0–14.1 | 14.2–25.2 | 25.3–36.2 | 36.3–47.3 | 47.4–58.4 |
| GWC (33 kPa) | 4.36–14.41 | 14.42–24.46 | 24.47–34.51 | 34.52–44.56 | 44.57–54.61 |
| GWC (1500 kPa) | 0.96–7.21 | 7.22–13.46 | 13.47–19.71 | 19.72–25.96 | 25.97–32.21 |
| COLE | 0.000–0.052 | 0.053–0.104 | 0.105–0.156 | 0.157–0.208 | 0.209–0.260 |
| pH | 4.3–5.0 | 5.1–5.7 | 5.8–6.5 | 6.6–7.2 | 7.3–7.9 |
| CEC | 1.74–10.60 | 10.61–19.46 | 19.47–28.33 | 28.34–37.19 | 37.20–46.05 |
| CaCO$_3$ | 0.00–4.83 | 4.84–9.67 | 9.68–14.50 | 14.51–19.34 | 19.35–24.17 |

### 3.2.1. Physical Properties

Texture is closely related to porosity, aeration, water retention, permeability, with the processes of waterlogging, hydromorphy, washing of salts and nutrients, etc., soil properties, and processes that have an enormous influence on the development of agricultural crops. Figure 2 shows the distribution of the main physical properties of the soil in the studied sector.

Figure 2a shows the distribution of total sand content. Moderate and high ranges predominate (sand between 40–80%), which translates into a predominance of loamy textures, characteristic of superficial horizons developed on sandstones. Loam soils have a "balanced" texture, i.e., the ideal texture for good crop development. On the other hand, it is worth noting the high content in the extreme northwest, which is related to the presence of sandy soils developed from weathered granites. Sandy soils have good aeration and are easy to till, but have low cation exchange capacity, are deficient in plant nutrients, have low water retention (they dry out very easily), and are very permeable (bases are easily washed out and transported out of the soil profile).

Figure 2b shows the clay distribution. In general, the predominant clay content is moderate. From the center to the northeast, there are soils with high clay content corresponding to the Luvisols typical of the region of La Armuña, while to the northwest, the clay content is minimal due to the presence of soils developed on granitic rocks. Regarding the south, the clay content in the analyzed profiles presents a low to moderate content, corresponding to soils formed on slate, in which textures with a high silt content predominate. In areas where the clay content is high, the storage of water and nutrients is high, which will have a positive effect on agricultural activities, which is enhanced by the presence in some areas of argillic subsurface horizons with clay illuviation (Bt). However, soils with an excess of clay (>40%) are impermeable, agricultural work becomes very difficult due to their strong plasticity in the wet state or excessive compaction in the dry state.

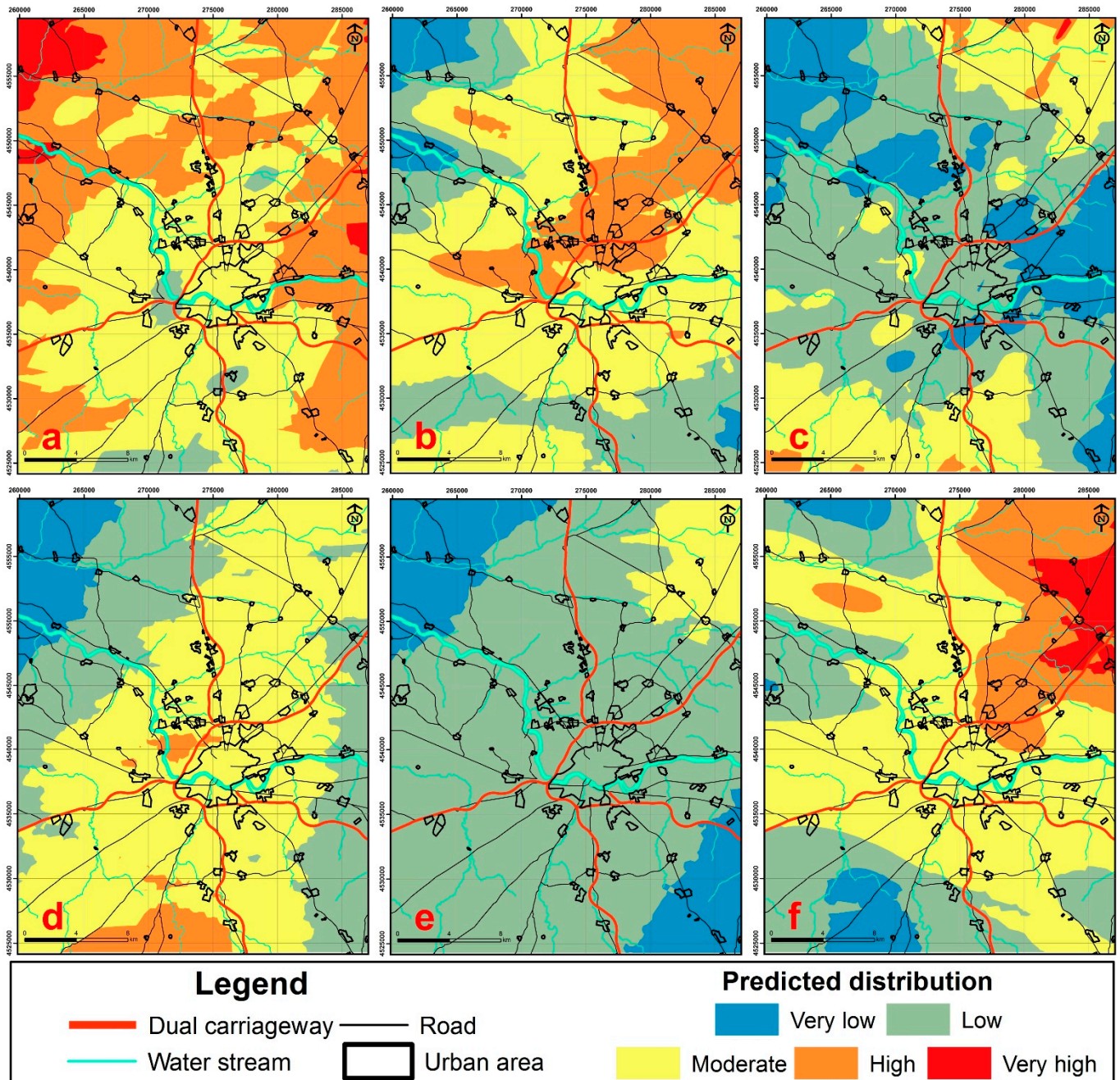

**Figure 2.** Distribution of the physical properties with the greatest impact on agricultural productivity in the soils studied:
(**a**) Sand; (**b**) Clay; (**c**) Organic carbon; (**d**) Water retention at 33 kPa; (**e**) Water retention at 1500 kPa; (**f**) COLE.

The organic carbon content is presented in Figure 2c. The organic carbon content in the study area is very low (<1%) for the most part, presenting a mean value of 1.5%. The distribution is very irregular and does not seem to follow clear patterns. This is because the main factor dominating the distribution of CO is the use to which the soil is subjected. The surface horizons of soils located in areas of natural vegetation (holm oak groves and pastures) have a double content (1.8%) in organic carbon than the Ap horizons of cultivated soils (0.8%), which shows the rapid degradation of organic matter in the surface horizons due to the effect of agricultural work. At present, in the face of climate change, arable soils generally have low organic carbon values, while values are higher under permanent vegetation cover. The conversion of natural land to cropland is one of the largest anthropogenic sources of carbon emissions and has led to the release of

about 200 Pg C over the last 250 years worldwide [56,57]. It has recently been recognized worldwide that soil carbon sequestration can be of great importance as a climate change mitigation and adaptation measure. Faced with this new challenge, the agricultural soils of the region studied could act as potential carbon sinks. To this end, it is necessary to promote agricultural techniques that favor the conservation and increase of carbon, such as: conservation tillage, addition of exogenous organic matter (manure, compost, etc.) and cover crops and fallows with vegetation.

The physical properties of the soil inform us about the capacity of the soil to provide a suitable physical environment for crop root growth. For this purpose, the soil must present low compaction so that it does not oppose excessive mechanical resistance to root advance; as well as a porosity that facilitates drainage, in wet periods, and the storage or retention of water to cover the needs of the plant, in dry periods. The distribution of values for soil water retention at 33 and 1500 kPa ($GWC_{33kPa}$ and $GWC_{1500kPa}$) are shown, respectively, in Figure 2d,e. The distribution of both properties is similar, however, the values are higher for the case of the $GWC_{33kPa}$, where moderate values predominate versus low values in the $GWC_{1500kPa}$. The minimum contents stand out in both cases in the extreme northwest, due to the sandy character of the soils in this granitic sector, since retention depends largely on the fine particle (clay) content of the soil. The soils with the highest water retention are the Vertisols, developed on loams and clays, and those with the lowest values are the Fluvisols, formed on sands. The use of crops with drought-resistant varieties is recommended in low GWC sectors.

Certain soils have the capacity to expand significantly when wet and to contract when dry, which is related to a relatively high content of montmorillonite type clays (smectites). This ability to expand and contract is quantified by using a coefficient called linear extensibility coefficient or COLE. Figure 2e shows the distribution of COLE values in the soils analyzed, ranging from 0.000 to 0.260. COLE is closely related to the clay content of the soils, since it assesses their expansibility or swelling. Moderate distribution predominates, being lower towards the south and northwest (presence of slate and granitic metasedimentary materials). The values are higher towards the northeast, coinciding with the clay-rich soils (Vertisols) of the region of La Armuña. Fluvisols and Leptosols are the soils with the lowest values. Soils with high COLE are very appropriate for the development of some leguminous plants (lentils) but can negatively affect other crops with low response capacity to conditions of excessive plasticity and dryness, during the winter and summer periods, respectively, and to the rupture of the root system of the plants, due to the processes of contraction and expansion of the clays.

### 3.2.2. Chemical Properties

The distribution of the main chemical properties of the soils of Salamanca and surrounding areas is presented in Figure 3. The cation exchange capacity (CEC) is a property by which anions or cations in the soil water can be exchanged with the anions or cations contained in the clay minerals (phyllosilicates) and organic matter, with which it is in contact. When an aqueous solution is brought into contact with certain soil substances, an exchange of ions takes place between the solid and the solution. The higher the CEC, the higher the fertility of the soil. The CEC ranges from 1.74 to 46.05 (cmol (+) $kg^{-1}$), the average value being 11.82 cmol (+) $kg^{-1}$ (Figure 3a). Values in the sector are irregular, with moderate-high values in the more clayey soils (Vertisols) and low values in the more sandy textured soils (Leptosols and Fluvisols). An increase in the organic matter content of the soil would lead to higher CEC values and thus improve crop yields.

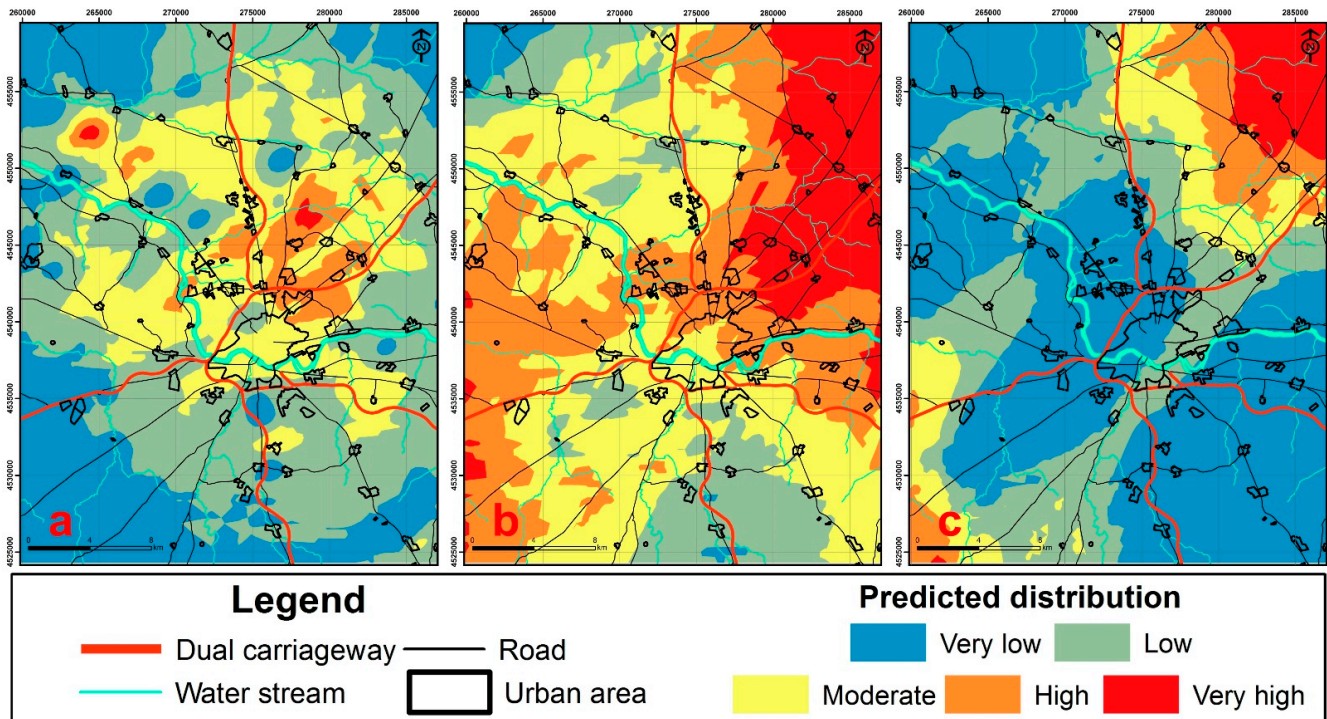

**Figure 3.** Distribution of the chemical properties with the greatest impact on agricultural productivity in the studied soils: (**a**) CEC; (**b**) pH; (**c**) CaCO$_3$.

The pH is a decisive factor in soil fertility since it can cause the immobilization (precipitation) of certain nutrients for plants; that is to say, at certain pH, the nutrients are not assimilated by plants (they are precipitated and are not soluble). On the other hand, at other pHs, these same nutrients are solubilized and assimilated. A low pH leads to a lack of nutrient availability, especially Ca and Mg [58,59]. It can be stated that a soil with intermediate pH (between 6.6 and 7.3) has the best conditions for most crops to assimilate soil nutrients. Figure 3b shows the pH distribution, with moderate and high pH values predominating. The highest values appear in the northeast and southwest quadrants, coinciding with the presence of highly developed soils with carbonate accumulations. The most acid soils are restricted to thin soils developed on geological materials of marked acid character: granites to the northwest, and slates to the south. The average pH value of the soils in the sector studied is slightly acidic (5.9), although there are actually two completely different populations of soils: acidic soils and calcareous or basic soils. The soil units with the highest pH correspond to the Calcisols and Vertisols. To produce an increase in crop yields, efforts must be made to increase pH, which can be addressed through the process of liming soils.

The carbonate content of soils is closely related to their pH (Figure 3c). The percentage of calcium carbonate equivalent of the samples studied ranges from 0.00 to 24.17%, with an average value of 1.37%. It is observed that low and very low values predominate in a large part of the study sector, and that the highest values correspond to the northeast and southwest. It should be noted that in the study area, the presence of carbonates is usually linked to washing and accumulation processes in the deep horizons of the soil, so that the presence in the superficial horizon is usually low. In the region studied, calcareous soils predominate over non-calcareous soils. Calcareous Regosols and Phaeozems and Calcisols are the soils with the highest percentages of carbonates. Soils developed on shales, quartzites, siliceous sandstones, etc. (Leptosols, Nitosols, and Acrisols) have negligible amounts of calcium carbonate equivalent.

### 3.2.3. Soil Nutrients

Properties related to soil chemical fertility refer to the soil's capacity to retain and supply the necessary nutrients for crop needs, such as nitrogen, phosphorus, and potassium (N, P, K). In a conventional agricultural system, nutrients come both from the mineralization of organic matter and weathering of minerals, as well as from inputs such as synthetic fertilizers. Regarding the fertility observed in soils, the distribution of the main nutrients is shown in Figure 4.

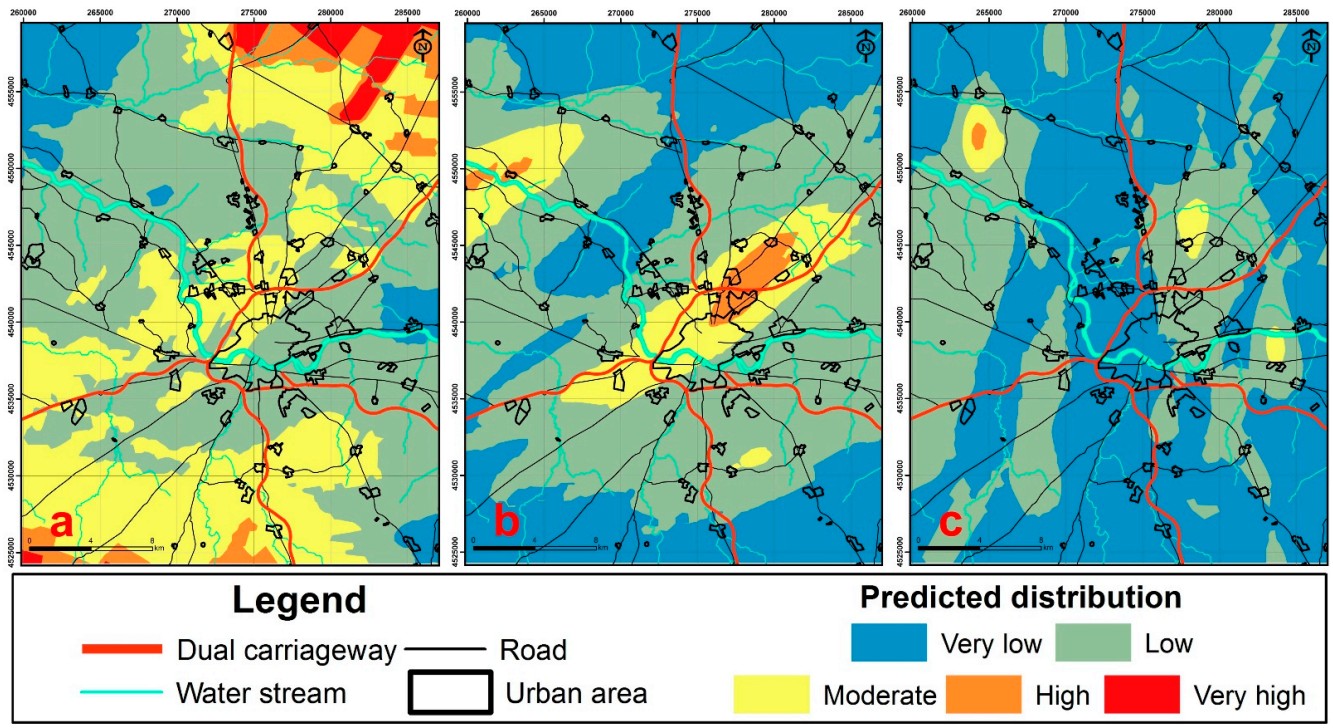

**Figure 4.** Distribution of the main soil nutrients in the study area: (**a**) N; (**b**) P; (**c**) K.

A productive soil must contain all the essential nutrients for plants in sufficient and proportional quantities. They must also be in an assimilable form so that plants can utilize them. The nutrients that have traditionally been studied in fertility programs are nitrogen, phosphorus, and potassium. They are used by plants in large quantities and are therefore referred to as "main or primary nutrients". The predominant nitrogen content (Figure 4a) is low at the regional level, which is clearly observed in large areas, reaching very low values in the poorer granitic soils. In the interior of the studied area, small areas with moderate N contents are also observed. Normally, N has a greater effect on crop growth, quality, and yield. However, N was deficient in most areas with values between 0.01 and 0.61% (low and very low). Acute N deficiency was due to low CO content, higher mineralization rate, and insufficient N fertilizer application to nutrient-depleting crops such as wheat. The rate of CO decomposition and soil N mineralization has complex interactions with the microbial population and other environmental factors, mainly soil moisture and temperature.

The assimilable phosphorus content ranges from 0.50 to 15.2 mg 100 $g^{-1}$, the average value being 4.23 mg 100 $g^{-1}$. It shows an irregular distribution characterized by two nuclei of moderate and high contents in the central and western sectors, with low and very low values predominating at the regional level. It is complex to establish distribution relationships for phosphorus. This usually comes from the parent rock, or it can be applied in phosphate fertilizers. In general terms, we can say that the soils studied have low phosphorus content, and that this can be a limiting factor for agricultural development if appropriate management measures are not taken. Phosphorus is more directly affected by soil pH than other major plant nutrients such as N and K. For example, at alkaline values,

the pH of the soil is higher than the pH of the soil. For example, at alkaline values greater than pH 7.5, phosphate tends to react rapidly with calcium (Ca) and magnesium (Mg) to form less soluble compounds. At acidic pH values, phosphate ions react with aluminum (Al) and iron (Fe) to form less soluble compounds again. Soils with pH values between 6 and 7.5 are ideal for the availability of P. In addition to pH, the amount of CO and the supply of phosphate fertilizers also control the availability of phosphorus in the soil, while erosion and runoff are associated with its loss.

As in other parameters related to soil fertility, the study area presents distributions dominated by low and very low potassium (K) contents (Figure 4c). Only in isolated areas are the concentrations of this element higher. The percentage of assimilable potassium is between 3.00 and 58.4 mg 100 $g^{-1}$, the average value being 10.06 mg 100 $g^{-1}$. Agricultural practices may be the cause of this "stress" suffered by the nutrients in the soils of Salamanca and surrounding areas. Soil pH affects the availability of potassium in the soil. When the pH is greater than 7, the higher Ca concentration increases K availability through the displacement of exchangeable K by Ca. Conversely, when soil pH is lower than 5.5, the reduction in Ca concentration reduces K availability. In addition, low CO levels, low clay content, and possible nutrient losses through leaching and erosion also reduce K levels.

### 3.2.4. General Agricultural Recommendations

This region is dominated by plains carved out of Cenozoic sediments, with the presence of powerful and fertile soils, which have allowed the establishment of high-yield rain-fed agriculture. The best soils in this region are considered to be deep soils (Luvisols), with a loamy texture in the surface horizon and a low organic matter content due to degradation by cultivation over several centuries, with a subsurface horizon of argillic type (Bt), in many cases with clays of the smectite group, with high water retention (the soils have a xeric moisture regime, which means a deficit in the water balance in the soil between April and October), with the presence of a horizon of carbonate accumulation and a pH close to neutral (the optimum for most crops), with a high capacity for cation exchange and a high degree of base saturation. On the contrary, in the northern and southern limits of the studied area, a steeper landscape can be observed, developed on the Palaeozoic base, with thin soils (Leptosols and Regosols), with a lithic contact near the surface that prevents root development, with a high degree of erosion and low fertility, which have traditionally lent themselves to a land use dedicated to extensive livestock farming (pastures). It is interesting to note that in this region, the most fertile soils have been used, as is logical for agricultural use for several thousand years; on the contrary, shallow soils, with abundant rockiness or stoniness, with a very sandy texture, low CEC, etc., have been used solely and exclusively for pasture [26,27].

The bad practices, so widespread in conventional agricultural soil management, lead to the major environmental problems described above. An alternative to the traditional system to avoid these problems would be the application of conservation and ecological soil management techniques. These practices have in common a more efficient supply of organic matter (application of organic amendments such as manure, compost, biochar), lower tillage intensity (conservation tillage), use of higher yielding species/biomass, or limitation in the use of agrochemicals. Such practices increase soil quality, improve soil fertility, retain more water, and reduce susceptibility to compaction and erosion.

### 3.3. Geostatistics for Agricultural Land Management

The semivariogram model and the main geostatistical parameters of the soil properties studied are shown in Table 4. The semivariogram model for each property was chosen based on the one with the lowest RMSE [60]. For the parameters GWC$_{33kPa}$, GWC$_{1500kPa}$, pH, and COLE, the exponential model provided the best fit to the semivariogram, with the circular model being the best fit to the semivariogram for the parameters N, K, and CEC, while the Gaussian was the best fit for the rest. In relation to the Sp. D of the soil parameters, it ranged from 0.02 (in CEC) to 0.91 (in GWC$_{33kPa}$), being high (in CEC), moderate (in Sand,

Clay, OC, N, P, K, GWC$_{1500kPa}$, COLE, pH, and CaCO$_3$), or weak (in GWC$_{33kPa}$). The ranges of spatial dependencies were large and typically vary between 2947 m for N and 10,565 m for P, indicating that the optimal sampling interval varies greatly between different soil properties. An abnormally high value was obtained for the water retention parameters GWC with a value of 44,607 m. As for the nugget effect, the highest value observed corresponds to CaCO$_3$ (7.26), which indicates discontinuity between samples [61] and is in line with it being the parameter with the worst fit, while the lowest value resulted for pH and CEC (0.02 and 0.04, respectively).

**Table 4.** Semivariance analysis of spatial structure in soil parameters.

| Parameters | Model | Range | Lag Size | Nugget | Partial | Sill | Nugget/Sill | Sp. D |
|---|---|---|---|---|---|---|---|---|
| Sand | G | 4236 | 353.0 | 0.21 | 0.16 | 0.37 | 0.57 | M |
| Clay | G | 6510 | 542.5 | 0.33 | 0.22 | 0.55 | 0.60 | M |
| OC | G | 3001 | 250.0 | 0.16 | 0.29 | 0.45 | 0.36 | M |
| N | C | 2947 | 245.5 | 0.24 | 0.18 | 0.42 | 0.57 | M |
| P | G | 10,565 | 880.4 | 0.48 | 0.30 | 0.78 | 0.62 | M |
| K | C | 4639 | 386.6 | 0.12 | 0.25 | 0.37 | 0.32 | M |
| GWC (33 kPa) | E | 44,607 | 3717.3 | 0.20 | 0.02 | 0.22 | 0.91 | L |
| GWC (1500 kPa) | E | 44,607 | 3717.3 | 0.29 | 0.10 | 0.39 | 0.74 | M |
| COLE | E | 8918 | 743.2 | 0.88 | 0.81 | 1.69 | 0.52 | M |
| pH | E | 3730 | 310.9 | 0.01 | 0.00 | 0.02 | 0.74 | M |
| CEC | C | 2982 | 248.5 | 0.04 | 1.63 | 1.67 | 0.02 | H |
| CaCO$_3$ | G | 8811 | 734.3 | 7.26 | 2.61 | 9.87 | 0.74 | M |

The mapping of specific soil parameters using geostatistics can represent a valid, fast, and inexpensive tool that provides useful and accurate information to evaluate the characteristics of agricultural land. Based on this information, agricultural practices and policies can be reformulated at local and regional levels with a consequent increase in agricultural yields as well as soil protection and improvement.

*3.4. Validation of Results*

The results of the cross-validation are summarized in Table 5. The MSE obtained can be considered, globally, as acceptable, due to their closeness to 0. With respect to RMSS, parameters with values close to 1 such as sand, COLE, and pH (0.9 < RMSS < 1.1) present very good precision. The parameters clay, phosphorus, water retention at 1/3 atm, and calcium were obtained with high precision, since they present values close to 1 (0.8 < RMSS < 0.9 and 1.1 < RMSS < 1.2). The properties organic carbon, nitrogen, potassium, water retention at 15 atm, magnesium, and exchange capacity showed acceptable accuracies (0.5 < RMSS < 0.8 and 1.2 < RMSS < 1.6). Finally, carbonates show a very low precision (RMSS close to 0), which is related to the two clearly differentiated populations that exist in the soils: soils on acidic ancient materials do not present carbonates, while in the case of tertiary materials, the presence is usual. In addition, the RMSS value shows that the model has underestimated the values for the parameters clay, organic carbon, nitrogen, potassium, water retention at 1/3 and 15 atm, COLE, pH, conductivity, and bulk density (RMSSE > 1), while for the remaining ones, it has overestimated them, although these errors are not high due to their closeness to 1 (except for carbonates). Finally, the RSME, the most commonly used meter, refutes these good accuracies obtained for most of the parameters, due to the closeness of its value with respect to the standard deviation of each parameter, which is reflected in Table 2.

**Table 5.** Errors obtained from cross-validation to estimate the interpolation accuracy.

| Parameters | MSE | RMSE | RMSSE |
|---|---|---|---|
| Sand | 0.0266 | 18.8581 | 0.9640 |
| Clay | −0.1050 | 14.1105 | 1.1391 |
| Organic Carbon | −0.1376 | 1.3265 | 1.2204 |
| Nitrogen | −0.1236 | 0.1001 | 1.2902 |
| Available phosphorus | −0.0699 | 3.4521 | 0.8164 |
| Available potassium | −0.1036 | 8.3775 | 1.3638 |
| Water retention (33 kPa) | −0.0753 | 8.6836 | 1.1145 |
| Water retention (1500 kPa) | −0.1635 | 6.0435 | 1.5121 |
| COLE | −0.1384 | 0.0446 | 1.0133 |
| pH | −0.0031 | 0.8596 | 1.0221 |
| Cation exchange capacity | −0.0095 | 12.5947 | 0.6881 |
| CaCO3 equivalent | 0.0019 | 5.3464 | 0.0062 |

Therefore, the results derived from the cross-validation show good and adequate accuracies for all parameters, except for carbonates, which is derived from the two very different populations existing in the studied area.

## 4. Conclusions

Knowledge of the distributions of the main physico-chemical properties of soils in an environment is essential for proper agricultural management. The application of geographic information systems and geostatistical methods, including descriptive statistics and semivariogram analysis, allows to improve the description of the spatial variability of the physicochemical properties of the arable layer of the soil. Descriptive statistics provide insight into data distributions and lay the foundation for geostatistical analysis. Geostatistical interpolation identified that the best fit of the semivariogram for each soil property can be achieved by different models (exponential, spherical, circular, or Gaussian) and, in general, showed moderate spatial dependence. In addition, the accuracy of the results was determined by cross-validation, with the interpolations generally showing low errors. The kriging maps of the properties studied were effective in explaining the distribution of soil properties in unsampled locations based on sampled data. These maps will help farmers to make efficient management decisions based on adequate knowledge of existing agricultural soil conditions. Therefore, these results, together with the agricultural measures that could be proposed based on them, show that geostatistical analysis using the kriging technique is an effective predictive tool to explore the spatial variability of soil properties.

**Author Contributions:** Conceptualization, M.C. and A.M.-G.; methodology, M.C.; software, M.C. and L.M.; validation, M.C. and F.S.-F.; formal analysis, A.M.-G. and F.S.-F.; investigation, M.C. and F.S.-F.; resources, A.M.-G.; data curation, A.M.-G.; writing—original draft preparation, M.C.; writing—review and editing, M.C. and A.M.-G.; visualization, L.M.; supervision, F.S.-F.; project administration, A.M.-G.; funding acquisition, A.M.-G. All authors have read and agreed to the published version of the manuscript.

**Funding:** This research received no external funding.

**Acknowledgments:** This research was supported by the project SA-044G18 of Regional Government of Castilla y Leon, and the GEAPAGE research group (Environmental Geomorphology and Geological Heritage) of the University of Salamanca.

**Conflicts of Interest:** The authors declare no conflict of interest.

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
