# Peer review of "Improving the Management of a Semi-Arid Agricultural Ecosystem through Digital Mapping of Soil Properties: The Case of Salamanca (Spain)"

_agronomy, doi:10.3390/agronomy11061189_

Round 1
Reviewer 1 Report
The manuscript present interesting results concerning the soil mapping based in soil physical and chemical characteristics, a technique that can help farmers to use the adequate agricultural management and to protect soils degradation. Likewise, the authors highlights in their study the efficacy of the soil mapping using geostatistical (kriging technique) to explore the spatial variability of soil properties. The subject of this work is interesting. Following, I have included some comments:
- The legend of figure 2,3,4 is too small, improve legend character of all figures.
- In results and discussion, the explication of the different soil parameters is in one block together, and it is too long. It is better to divided the results is session like soil texture, Soil chemical fertility (N, P, K), Soil Ph and cation exchanges, Cole and water retention, it is better to explain each figure in a separate session, with this manner results are more readable.
- In their objectives, authors indicated 3) to establish adequate management guidelines that help local farmers to improve agricultural yields in their crops. It would be perfect, if the authors provide a summary or a recapitulate table (depending on the results obtained in this study) of the adequate agricultural practices of each area studied, soil needs of fertilization or irrigation,,,,,,,,. Recommendation that help farmers and contribute to precision agriculture and soil sustainability. Finally, I consider this work very interesting, explaining soil properties using geostatistical method and I think that it is very important that authors project this results in farmers agricultural practice.
Author Response
Thank you very much for your contributions to improve this paper.
Please see attached file
We have improved the paper according to your indications. We hope that in the present version it will be ready for publication.

Reviewer 2 Report
See attached file

Author Response

(The authors gave the same response as above.)
